# An Easy and Ecological Method of Obtaining Hydrated and Non-Crystalline WO_3−x_ for Application in Supercapacitors

**DOI:** 10.3390/ma13081925

**Published:** 2020-04-19

**Authors:** Mariusz Szkoda, Zuzanna Zarach, Konrad Trzciński, Grzegorz Trykowski, Andrzej P. Nowak

**Affiliations:** 1Department of Chemistry and Technology of Functional Materials, Faculty of Chemistry, Gdańsk University of Technology, Narutowicza 11/12, 80-233 Gdańsk, Poland; zuziaz696@gmail.com (Z.Z.); trzcinskikonrad@gmail.com (K.T.); andnowak@pg.edu.pl (A.P.N.); 2Faculty of Chemistry, Nicolaus Copernicus University in Toruń, Gagarina 7, 87-100 Toruń, Poland; tryki@umk.pl

**Keywords:** exfoliation, hydrated non-crystalline WO_3_, supercapacitors, lithium-ion batteries, anode material

## Abstract

In this work, we report the synthesis of hydrated and non-crystalline WO_3_ flakes (WO_3−x_) via an environmentally friendly and facile water-based strategy. This method is described, in the literature, as exfoliation, however, based on the results obtained, we cannot say unequivocally that we have obtained an exfoliated material. Nevertheless, the proposed modification procedure clearly affects the morphology of WO_3_ and leads to loss of crystallinity of the material. TEM techniques confirmed that the process leads to the formation of WO_3_ flakes of a few nanometers in thickness. X-ray diffractograms affirmed the poor crystallinity of the flakes, while spectroscopic methods showed that the materials after exfoliation were abundant with the surface groups. The thin film of hydrated and non-crystalline WO_3_ exhibits a seven times higher specific capacitance (C_s_) in an aqueous electrolyte than bulk WO_3_ and shows an outstanding long-term cycling stability with a capacitance retention of 92% after 1000 chronopotentiometric cycles in the three-electrode system. In the two-electrode system, hydrated WO_3−x_ shows a C_s_ of 122 F g^−1^ at a current density of 0.5 A g^−1^. The developed supercapacitor shows an energy density of 60 Whkg^−1^ and power density of 803 Wkg^−1^ with a decrease of 16% in C_sp_ after 10,000 cycles. On the other hand, WO_3−x_ is characterized by inferior properties as an anode material in lithium-ion batteries compared to bulk WO_3_. Lithium ions intercalate into a WO_3_ crystal framework and occupy trigonal cavity sites during the electrochemical polarization. If there is no regular layer structure, as in the case of the hydrated and non-crystalline WO_3_, the insertion of lithium ions between WO_3_ layers is not possible. Thus, in the case of a non-aqueous electrolyte, the specific capacity of the hydrated and non-crystalline WO_3_ electrode material is much lower in comparison with the specific capacity of the bulk WO_3_-based anode material.

## 1. Introduction

The increase in global energy consumption, caused by the rapid increase in population, forces the production of energy storage and energy conversion systems on a larger scale. However, the utilization of fossil fuels has led to changes in the Earth’s climate by increased emissions of greenhouse gases [1]. Thus, new energy sources are expected to meet the requirements of being renewable and emission-free [2]. To meet these expectations, more and more attention is being paid to energy storage devices such as supercapacitors and batteries. Supercapacitors may store electrical energy of a high power density but low energy density without any conversion reactions [3]. Batteries require an electrochemical reaction to obtain electrical energy from a chemical reaction in a reversible way [4], assuming a high energy density. Despite the various energy storage mechanisms, one can assume that transition metal oxides may be successfully adapted as electrode materials for supercapacitors as well as for batteries (for example, Co_3_O_4_, MnO_x_, TiO_2_) [5,6,7]. This is due to the fact that properties of metal oxides are strongly dependent on many factors, i.e., crystal structure, morphology, doping and oxygen deficiency [8]. One of the methods of some transition metal oxides’ modification is exfoliation, which leads to the formation of 2D flakes.

The amusing physical and structural properties of atomically thin 2D materials are the result of their improved surface-to-volume ratio, altered surface energy and the confinement effect [9,10]. Throughout recent years, rapid progress has been achieved in establishing procedures of graphene preparation [11] and transition metal dichalcogenides in the form of single or few layers [12,13,14], as a consequence of the slow breakdown of van der Waals force between the adjacent layers in bulk. Nevertheless, the family of 2D materials is being expanded, especially by transition metal oxides, in order to greatly develop their progressive applications [15].

Among many transition metal oxides, tungsten (VI) oxide (WO_3_) is attracting much attention. It has been used in electrochromic devices [16], gas sensors [17], electrocatalysis [18] and photoelectrocatalytic processes [19]. Recently, many studies have shown that WO_3_ may be utilized as an energy storage material [20,21,22]. Lokhande et al. evidenced that the crystal structure affects the energy storage ability of WO_3_-based electrode materials [23]. There are many factors of synthesis that have an influence on the crystallographic structure, morphology and properties of obtained materials, i.e., temperature, pH, pressure, the time, the presence of additives, etc. [24,25,26,27]. Exfoliation is one of the modification methods to obtain 2D nanomaterials from layered materials [28] and is expected to improve electrodes’ performance in terms of both stability and capacity [29]. Apart from tungsten oxide, there is a major number of other metal oxides that were reported to enhance working properties of the energy storage devices. Ruthenium oxide is widely investigated because of its high specific capacitance (up to 700 F/g), but its application is severely limited by the high cost [30]. Metal oxides such as MnO_2_ [31] or NiO [32] have similar advantages and there have been some attempts to apply them as electrode materials. However, their poor electrical conductivity affects the speed of charging and contributes to severe capacitance loss. To improve the capacitive performance of materials with a poor electrical conductivity, many researchers found oxygen-defective metal oxides to be of the greatest interest. Reported results show that the concentration of oxygen vacancies has a significant influence on the structure, as well as charge storage properties, and thus enable an excellent cycling performance [33,34,35,36].

Several approaches are accessible for the manufacture of WO_3_ 2D materials, including hydrothermal, solvothermal, plasma treatment, acid etching, anodization and exfoliation. Over the last decade, many methods for the exfoliation of layered materials have been investigated for the synthesis of monolayer nanostructures. Mechanical exfoliation was firstly applied by Geim and Novoselov [37] to obtain graphene by using adhesive tapes. Nevertheless, it is difficult to obtain uniform samples by this technique. The other one, and one of the most commonly used techniques, is liquid exfoliation that could be combined with oxidation or ion intercalation/exchange, as well as surface passivation by solvents [10]. In the case of metal oxides, the exfoliation usually involves sonication in a surfactant solution and ion or polymer intercalation [38,39,40]. The application of the latter method leads to obtaining single layers in a crystalline form, possessing the promising properties and gives large quantities of a dispersed nanomaterial. However, amorphous materials also find various applications, yet they are still gaining less interest. To our knowledge, most of the research in which amorphous materials of a nanometric scale were obtained are limited in zero-dimensional (0D) and one-dimensional (1D), like amorphous MoO_3–x_ nanoparticles or amorphous FeCoPO_x_ nanowires [41,42,43,44,45]. All of the above-mentioned showed excellent activity in catalysis and local surface plasmon resonances (LSPR). However, there are still many attempts to obtain 2D materials, especially with confined thickness, that would have the superior activity and the ability of application in many fields. One of the latest studies, provided by Ren et al. [46], suggested using the supercritical CO_2_ technology to obtain the two-dimensional amorphous heterostructures of Ag/a-WO_3−x_ and proposed the concept of synergistic photocatalysis, which would serve as a new methodology for the design of a high-efficiency catalyst.

In this work, we used a process to modify the material, which is described in the literature as exfoliation. However, based on the results obtained, we cannot say unequivocally that we have obtained an exfoliated material. Nevertheless, we received a material characterized by non-crystallinity and the surface was enriched with the surface groups. Hence, this paper outlines the influence of a facile water-based strategy and its effect on the structure, morphology and electrochemical properties of WO_3_. It is evidenced that this strategy may improve the electrode’s performance for a supercapacitor application but it is not convenient for energy storage via a faradaic reaction in batteries.

Even if, for instance, the exfoliation process, proposed in the literature and used by us, leads to obtaining an exfoliated material, the presence of obtained structures would, in this work, result from the method of electrode preparation that consists of the slow evaporation of water which may partially reverse the exfoliation effect. Nevertheless, the proposed modification procedure clearly affects the morphology of WO_3_ and leads to the loss of crystallinity of the material.

## 2. Materials and Methods

### 2.1. Synthesis of Hydrated and Non-Crystalline WO_3_ and Electrodes Preparation

Bulk WO_3_ (Sigma, analytical grade, Saint Louis, MI, USA) was added, as received, to triple-distilled water to create a suspension with a concentration of 60 mg mL^−1^ and was refluxed for 10 days at 80 °C. The suspension obtained after 10 days was transparent and bulk WO_3_ residues could not be distinguished. Very often centrifugation is necessary to separate larger, non-exfoliated crystallites. In the case of this developed procedure, the step of centrifugation can be omitted because under the proposed conditions, the process occurs with almost 100% efficiency. The solid residues were not present in the exfoliated solution, as it is presented in Figure 1. The exfoliation of WO_3_ occurs readily, even under dark conditions (via covering the reaction vessel with aluminum foil), which reveals that the visible light does not play a significant role in the exfoliation process. In this paper, hydrated and non-crystalline WO_3_ was synthesized via refluxing under natural light.

In the case of the modified WO_3_, the working electrodes for the electrochemical measurements in an aqueous electrolyte were prepared by a drop-casting method. The volume of 200 μL of an aqueous suspension of the WO_3−x_ was drop-casted onto the ~0.67 cm^2^ degreased FTO (fluorine-doped tin oxide) glass (Sigma, 7 Ω/sq, Saint Louis, MI, USA). The mass loading of the electrode material after water evaporation at 80 °C ranged from 5.6 to 8.2 mg cm^−2^. WO_3_ bulk powder was deposited onto the FTO glass using the dip-coating method. At first, 0.2 g of material and about 0.1 g of poly(ethylene oxide-PEO) (M = 300,000, Aldrich) were mixed with 1 ml of water in order to obtain a homogenous mixture. The degreased FTO was immersed in the WO_3_/PEO/H_2_O, pulled out, dried, and heated for 5 h at 400 °C in an air atmosphere. The thermal treatment led to the PEO (poly(ethylene oxide)) removal from the deposited film. The procedure of solid material deposition on the substrate was previously elaborated [47]. In this case, the mass of the WO_3_ layer loading ranged from 7.2 to 11 mg cm^−2^.

The suspension concentration of the hydrated WO_3−x_ was transferred to the Petri dish. The excess of water was slowly evaporated at 45 °C. The obtained yellow powder of WO_3−x_ nanoflakes was dried at 80 °C for 24 h and was used for the material characterization (TEM, SEM, XRD, XPS, FTiR), as well as for the electrode preparation for the electrochemical measurements performed in a non-aqueous electrolyte. Battery tests were performed with electrodes obtained from a slurry containing WO_3_ (bulk or modified), carbon black (CB) (Super P, Timcal, Bodio, Switzerland), and pVDF (Polyvinylidene fluoride, Solvay) at a weight ratio 7:2:1 dissolved in NMP (N-methyl-2-pyrrolidinone, AlfaAesar) on a copper current collector (Schlenk Metallfolien GmbH & Co KG, Georgensgmünd, Germany). The discs of 10 mm were cut off, pressed for 30 s under a 2 MPa load, and finally dried for 24 h at 100 °C under a dynamic vacuum in an oven (Glass Oven B-595, Büchi, Germany). The mass loading of the electrode material was around 3 mg cm^−2^.

#### 2.1.1. Morphology and Crystal Structure

The morphology was examined using scanning electron microscopy (Quanta 3D FEG, Fei Company), with the beam accelerating voltage kept at 20 kV, and transmission electron microscopy (Tecnai 20F X-Twin, Fei Company, Cambridge, UK). The preparation of the samples for the TEM imaging consisted of sonication for 5 s of a few milligrams of bulk or modified WO_3_ in ethanol (99.8% anhydrous) using ultrasounds, and then an applied drop (5 µL) on a carbon-coated copper mesh with holes (Lacey type Cu 400 mesh, Plano), and the evaporation of the solvent was at a room temperature. The crystal structure of the prepared materials was studied with powder X-ray diffraction (XRD). Patterns were obtained by an X‘Pert Pro diffractometer with an X‘Celerator detector and Cu Kα radiation, λ = 0.15406 nm. The infrared spectra (FTiR) in the 300–4000 cm^−1^ range were recorded in a vacuum spectrometer Vertex 70V, Bruker Optic, with a T = 22 °C, p = 10^−1^ Pa, resolution of 4 cm^−1^, and a number of 50 scans. The sample was mixed with KBr at a mass ratio of 1/300, then compressed at 7 MPa to form a pellet, and then the transmission spectrum was recorded. The chemical composition measurements for the material after the exfoliation process were performed by the X-ray photoemission spectroscopy method. The XPS measurements were performed using an Argus Omicron NanoTechnology X-ray photoelectron spectrometer.

#### 2.1.2. Electrochemical Measurements

Aqueous electrolyte: The electrochemical measurements were performed in a 0.2 M K_2_SO_4_ aqueous electrolyte purged with argon. A three-electrode cell was used for the cyclic voltammetry and galvanostatic charge–discharge cycles measurements. A platinum mesh acted as a counter electrode and Ag/AgCl (3 M KCl) was used as the reference electrode. The electrochemical studies were conducted using the AutoLab PGStat204 potentiostat-galvanostat system (Methrom, AutoLab, Utrecht, The Netherlands). The charge–discharge measurements were carried out with a current density equal to 0.2 A g^−1^ in a polarization range from −0.15 to 0.9 V vs. Ag/AgCl (3 M KCl). The galvanostatic charge–discharge measurements for the hydrated WO_3−x_ were also performed using fully assembled symmetric two-electrode cells in a coffee bag system. The commercially available foil was used for the preparation of the supercapacitor cells. A Whatman paper was used as a separator. An aqueous solution of 0.2 M K_2_SO_4_ was used as an electrolyte. Coffee bags were enclosed under a vacuum using a Mini Jumbo Henkelman Vacuum System. The supercapacitor cells were tested using multiple galvanostatic charge–discharge cycles (10,000 cycles, j_c_ = j_a_ = 0.5 A g^−1^).

The electrochemical impedance spectra (EIS) for both electrode materials were recorded using AutoLab PGStat10 for the working electrode at its rest potential and under anodic polarization. The frequency range covered 10 kHz–0.32 Hz (90 points), whereas the amplitude of the AC signal equaled 10 mV. The following elements were used for the fitting procedure of the measured impedance spectra using EIS Analyzer software: *R*: resistance, *CPE*: constant phase element, and *Z*_Wo_: finite length diffusion impedance, where:(1)$$Z_{W_{o}}\left(\omega \right)=\frac{W_{or}}{\sqrt{\omega }}\left(1-j\right)\coth\left(W_{oc}\sqrt{j\omega }\right)$$
(2)$$Z_{CPE}\left(\omega \right)=P^{-1}{\left(j\omega \right)}^{-n}$$

Non-aqueous electrolyte: The electrode materials were tested in two-electrode Swagelok^®^ cell with lithium foil (99.9%, 0.75 mm thickness, AlfaAesar, Haverhill, MA, USA) as a counter and a reference electrode. The SelectiLyte™ LP30 (1 M LiPF6 in EC/DMC 50:50, wt %) from Merck was used as an electrolyte and a glass fiber (Schleicher & Schüll, city, Germany) as a separator. The battery tests of the samples were performed using the ATLAS 0961 MBI (ATLAS_SOLLICH, Banino Poland) multichannel battery testing system at current densities of 50 mA g^−1^, from 0.01 V to 3 V. The cyclic voltammetry measurements (CV) were carried out on a PGStat204 galvanostat/potentiostat over the potential range from 0.005 V to 3V vs. Li/Li^+^, with a scanning rate of 0.1 mV s^−1^.

## 3. Results and Discussion

### 3.1. Morphology and Structure

The morphology of the WO_3_ samples was tested using the SEM technique. The micrographs of the bulk WO_3_ are presented in Figure 2a,b. As can be seen, the non-modified powder is built of grains of various sizes in the order of several dozen micrometers. Regular shapes and sharp edges are distinctive for a crystalline material. Higher magnification reveals a layered structure of the bulk WO_3_ (see Figure 2b). The material after modification is characterized by a different morphology, as is shown in Figure 2c. Slowly-dried, hydrated WO_3−x_ forms agglomerates with irregular shapes. They are clearly different to the crystallites of the bulk WO_3_. Aggregates of an exfoliated WO_3_ are built of randomly oriented plates of heterogeneous shapes with a size of 0.7–1.5 μm (see Figure 2d and inset Figure 2d). The thicknesses of the flakes were roughly estimated to be in the order of 50–100 nm (see Appendix A). The presence of such thick structures results from the method of sample preparation that consists of the slow evaporation of water which may partially reverse the exfoliation effect. Thus, the proposed modification procedure clearly affects the morphology of WO_3_ and leads to the delamination of the material. However, a better insight into the morphology of the material was obtained using TEM. As is shown in Figure 3, the appropriate sample preparation for the TEM imaging allows ultra-thin nanoflakes of WO_3−x_ to be observed. The transparency of the layers under TEM reveals that the thickness should not exceed a few nanometers.

FTIR spectroscopy was used to analyze the functional groups of the bulk WO_3_ and WO_3_ after modification. As can be seen in Figure 4, the two spectra are different from each other in the range 600–1000 cm^−1^. In the case of the modified WO_3_, peaks are observed at 780, 890 and 940 cm^−1^. These bands can be attributed to the O-W-O vibrations, while the band at 1630 cm^−1^ is linked to the bending modes of the O–H groups. The W-O stretching modes are less intense, and changes in the low-frequency modes may indicate some modifications in the tungsten oxide framework. The appearance of the most active surface centers suggests a connection with defects in the nanoflakes [48]. The non-crystalline WO_3_ sample has a large number of defects that somehow activate W-O vibrations in the range 600–1000 cm^−1^. This phenomenon is missing in the bulk sample, which is why the W-O vibrations are inactive, and thus not visible in the FTIR spectrum are peaks at 780, 890 and 940 cm^−1^. A similar effect was observed on the pristine Zn-Al and exfoliated samples [49]. Another band at 3430 cm^−1^ is linked to the stretching modes of the OH groups in water or hydroxyls [50]. On the basis of relative intensities, it can be concluded that there are many more OH groups in the modified sample. It can also be indicated by the widening of the peak at 3400 cm^−1^, that can be divided into two bands at approximately 3430 and 3100 cm^−1^. Both bands are assigned to the OH group, however, they differ chemically. The OH group at 3430 cm^−1^ is connected to the adsorbed water, and the OH group at 3100 cm^−1^ is directly bound to the WO_3_ structure.

The crystal structures of the samples were examined by the powder X-ray diffraction technique (Figure 5). In the case of the bulk WO_3_, all the reflexes could be assigned to the monoclinic phase of WO_3_. No other reflexes were detected, confirming phase purity. As is shown in Figure 5a, the amorphous structure was detected for the WO_3−x_ nanoflakes, suggesting the lack of crystallinity. Poor crystallinity in the exfoliated transition metal oxide (MoO_3−x_) was also observed by Ahmed S. Etman et al. [51]. Nevertheless, low-intensity peaks can be found (see Figure 5b). However, these signals do not come from the crystalline form of WO_3_. On the basis of the XRD pattern, it can be concluded that hydrated WO_3_ is composed of an amorphous phase, but both orthorhombic tungsten trioxide hydrate (o-WO_3_·H_2_O) and hexagonal tungsten oxide hydrate (h-WO_3_·0.33H_2_O) traces can be detected [52].

Due to the fact that the obtained result in the form of the hydrated and non-crystalline material was unexpected after applying the exfoliation technique, it is difficult to make a comparison as most of the exfoliation results provide a crystalline phase of the material. What is more, the process of the exfoliation of transition metal oxides, a tungsten oxide in particular, is not as widely studied as, for example, the process of graphene exfoliation. However, there are individual studies on this topic. In both Kalantar-Zadeh et al.’s [53] and Yan et al.’s [54] works, they used hydrated WO_3_ as the precursor, which is a typical layered material consisting of planes connected through the interaction of oxygen and hydrogen through hydrogen bonds in the adjacent layers. Waller et al. [55] exfoliated Bi_2_W_2_O_9,_ and in each of these researches, the crystalline phase of the final product was obtained. By comparing the results presented in Table 1, it can be stated that regardless of which exfoliation technique was used, tungsten oxide was obtained in a crystalline form. In the case of the nanoflakes obtained during the experiment presented in this work, the exfoliated product might have undergone hydration, which has determined its final properties. The use of the water exfoliation technique may also be related to obtaining poor crystallinity. Similar results were obtained by Etman et al. [51], who also observed poor crystallinity for the MoO_3_ nanosheets, also utilizing a water-based exfoliation.

X-ray photoelectron spectroscopy (XPS) analysis was employed to investigate the oxidation states and the presence of oxygen vacancies in the hydrated and non-crystalline WO_3_. The high-resolution XPS spectra for the tungsten and oxygen region are presented in Figure 6. The W 4f core-level XPS spectrum can be deconvoluted into four peaks corresponding to the W^6+^ and W^5+^ states (see Figure 6a). The peaks located at the binding energies of 35.83 and 37.97 eV correspond to the W^6+^ state, while the peaks at 34.96 and 37.11 eV correspond to W^5+^. The splitting between W 4f7/2 and W 4f5/2 for the W^6+^ state is 2.13 eV, which is in good agreement with an earlier report [58]. The emergence of the W^5+^ oxidation state can be associated with the presence of oxygen vacancies that are formed during exfoliation. Therefore, the chemical structure can be referred to as WO_3–x_.

The O 1s spectrum shows a broad asymmetric peak, which can be deconvoluted into three peaks, as shown in Figure 6b. The main peak (red curve) at 530.4 eV is assigned to the oxygen atoms (O^2−^) that form the strong W = O bonds [59,60]. This value is slightly lower than that reported for the commercial powder WO_3_ (531 eV) [61], suggesting a change in the coordination environment between the O and W atoms in the exfoliated WO_3_. The peak at 531.6 eV (blue curve) can be assigned to the surface-adsorbed species (OH^−^, O^−^, or oxygen vacancies) [62,63]. Finally, the peak at 533.4 eV (green curve) can be assigned to adsorbed water [63], proving the existence of WO_3_(H_2_O)_n_ phases at the surface [64].

### 3.2. Electrochemical Properties

#### 3.2.1. Aqueous Electrolyte

##### Three-Electrode System Configuration

Electrochemical studies of the modified WO_3_ electrodes were carried out in order to evaluate their utility as an electrode material for supercapacitors. The CV (cyclic voltammetry) curves recorded for the bulk and modified electrode materials are presented in Figure 7a. The measurements were carried out in the potential range of −0.15 to +0.9 V vs. Ag/AgCl/3M KCl, with a scan rate equal to 50 mV s^−1^. The hydrated WO_3−x_ material displays a highly rectangular shape without redox peaks, which indicates a capacitive nature of the electrode material. Cyclic voltammetry results show a good charge–discharge reversibility of the electrode process. The current plateau characteristic for electrical double layer capacitance (EDLC) was recorded. The EDL activity observed at a wide range of applied potential may be related to the presence of a high concentration of oxygen vacancies and OH surface groups, whose presence was confirmed using XPS and IR techniques. It was previously reported that WO_3_-based supercapacitors exhibited both pseudocapacitance and EDLC. However, the contribution of pseudocapacitance was much higher [20]. In the case of the hydrated WO_3−x_, the electrode material characterized by a specific morphology as well as having a surface enriched in surface groups, it may be expected that the contribution of EDLC is significantly higher in comparison with the bulk WO_3_. The contribution of pseudocapacitance is also expected, however, the lack of clear oxidation/reduction peaks on the CVs may be related to the lack of the hydrated WO_3−x_crystallinity. As can be seen, the bulk WO_3_ exhibited decidedly lower current densities compared with the non-crystalline WO_3_ nanoflakes. The CV curve of the bulk WO_3_, in contrast to the modified material, exhibits oxidation/reduction peaks that were observed previously and described as the conversion of the valence states of W centers with simultaneous adsorption/desorption of cations available in the electrolyte [65].

The CVs of the hydrated WO_3-x_ recorded at different scan rates (5–500 mV s^−1^) are presented in Appendix A. The anodic current density at 0.5 V shows a linear relation with the scan rate (see Appendix A), suggesting that the charge storage is controlled by the surface processes, thus it is not a diffusion-controlled phenomenon (see Appendix A).

Both electrode materials were compared using electrochemical impedance spectroscopy. Two spectra for each type of electrode were registered (Appendix A). The first one was recorded close to the rest potential (0 V vs. Ag/AgCl (3 M KCl), and the second under anodic polarization (0.6 V vs. Ag/AgCl (3 M KCl). The results, equivalent circuits, and fitting parameters are presented in the Appendix A (see Appendix A and Appendix A). The EIS analysis confirms that the diffusion process affects the impedance spectrum only of the bulk WO_3_ recorded at 0 V, seen as a straight line inclined at an angle of 45° at the lowest frequencies, and the Warburg element (W_o_) is necessary to fit the model properly. In the case of a hydrated WO_3−x_ model, it can be fitted using a simpler equivalent circuit, without the W_o_. The comparison of the fitting parameters of the bulk and hydrated tungsten oxide electrodes recorded under anodic polarization confirms that hydrated WO_3−x_ can act as an electrode material for energy storage devices, mainly supercapacitors.

Multiple galvanostatic charge–discharge tests were carried out for both electrodes in a 0.2 M K_2_SO_4_ aqueous electrolyte (Figure 7b). The polarization with an anodic and cathodic current gives almost identical capacitance values, indicating a reversible process with over 99% of columbic efficiency (Figure 7c). The electrode that contains the non-crystalline WO_3_ exhibits a seven times higher specific capacitance than the bulk WO_3_ due to the higher surface area of the interface between the electrode material and the electrolyte, and a higher concentration of W-OH surface groups that participate in the charge storage process. After a sequence of 1000 charge–discharge cycles, over 92% and 80% of the initial capacitances were maintained for the non-crystalline and bulk WO_3_, respectively. In the case of the bulk WO_3_, the capacitance retention after 1000 cycles was lower compared with the modified material. It may be related to the fact that the capacitance of the bulk WO_3_ mainly comes from the W centers reduction/oxidation and ions’ intercalation/deintercalation, while a modified material utilizes a highly reversible electroactivity of the surface groups.

##### Two-Electrode Configuration

Multiple charge–discharge cycles in a two-electrode configuration for the hydrated WO_3−x_ were performed in order to test the stability of the tested supercapacitor (see Figure 8). As is shown, a very good stability, even after 10,000 cycles, was obtained for the hydrated WO_3−x_-based capacitor. The capacitance retention between the 1st and the 10,000th chronopotentiometry cycle was equal to 84%. The effect of the capacitance drop was also tracked using chronopotentiometry. The curves recorded before and after a long-term test are presented also in the Figure 8 inset. It is noteworthy that the decrease of the capacitance is the highest at the beginning of the charge–discharge tests, and then the capacitance stabilized after approximately 2000 cycles. This means that the capacitance drop is not related to the electrolyte decomposition, but some irreversible reactions on the material surface.

The dependence of the specific capacitance of the hydrated WO_3−x_ electrode on the current density is shown in Figure 9a. The specific capacitance (C_s_) was 122 A g^−1^ and 72 A g^−1^ at a current density of 0.5 A g^−1^ and 4 A g^−1^, respectively. The capacitances were found to decrease by increasing the charge–discharge current. This is because, at a higher current density, the slower processes demonstrate a kinetic resistance and cannot participate in a charge transfer onto or across the electrode/electrolyte interface.

The Ragone plots display the relationship between the power density and energy density. In the case of the hydrated WO_3−x_, at a power density of 803 W kg^−1^, an energy density of 60 W h kg^−1^ was obtained. When the power density increased to 2520 W kg^−1^, the energy density was 36 W h kg^−1^ (see Figure 9b), which is quite impressive as compared with earlier reports about modified metal oxides (Appendix A) [32,33,35,66,67,68,69,70,71,72,73,74].

#### 3.2.2. Non-Aqueous Electrolyte

The electrochemical performances of the bulk and non-crystalline WO_3_ electrode materials in a non-aqueous electrolyte are shown in Figure 10a. The first cyclic voltammetry curve for the bulk WO_3_ exhibits cathodic peaks corresponding to the lithium insertion into the WO_3_ material (2.65 V and 2.37 V), according to the reaction below:WO_3_ + *x*Li^+^ + *x*e^−^↔Li_x_W_x_^(V)^W^(VI)^_x−1_O_3_ for 0 ≤ x ≤ 1(3)

The cathodic maximum at 0.62 V might originate from a conversion reaction as shown below:Li_x_W_x_^(V)^W^(VI)^_x-1_O_3_+ yLi^+^ + ye^−^↔ W + 3 Li_2_O for x + y = 6 and 0 ≤ x ≤ 1(4)

Reaction (3) is similar to the reaction proposed for photointercalated molybdenum bronzes [75]. However, in the literature, one may find that WO_3_ reacts with Li^+^ and gives Li_x_WO_3_ [76,77], instead of Li_x_W_x_^(V)^W^(VI)^_x−1_O_3_.

During the oxidation process of the bulk WO_3_, delithation occurs as a sharp anodic maximum at 1.25 V and a broad hump at 1.9 V. The former is attributed to the oxidation of W to Li_x_W_x_^(V)^W^(VI)^_x−1_O_3_, while the latter corresponds to the formation of WO_3_. The cathodic maxima for the second cycle did not overlap with the cathodic maxima from the first cycle. In the second cycle there is no cathodic maximum at 0.62 V. The lack of a cathodic maximum at 0.62 V may suggest that the conversion reaction of a monoclinic WO_3_ into Li_x_W_x_^(V)^W^(VI)^_x−1_O_3_ followed by the formation of a pure W is irreversible. This means that during the oxidation process, no monoclinic WO_3_ is formed as a final product. There are three cathodic maxima at 1.67 V, 1.2 V and 0.15 V. The first two maxima may confirm the formation of Li_x_W_x_^(V)^W^(VI)^_x−1_O_3_, while the last maximum might be attributed to the lithium ion intercalation into carbon black, which is one of the components of the electrode material. The cyclic voltammetry curves of the oxidation process of the bulk WO_3_ electrode material for the first and the second cycles are similar. One may see a broad anodic maximum at 1.3 V for both cycles. This maximum corresponds to the Li_x_W_x_^(V)^W^(VI)^_x−1_O_3_ oxidation with the WO_3_ formation.

In the case of the non-crystalline WO_3_, the shape of the first cyclic voltammetry curve differs from the shape of the first cyclic voltammetry curve originating from the bulk WO_3_ electrode material. One may see broad humps with cathodic maxima at 2.2 V, 1.67 V, 1.45 V, 1.04 V and 0.25 V, with one wide anodic maximum at 1.41 V. The difference in shape may origin form the lack of a crystallographic order of the WO_3_ nanoflakes. It is known that lithium ions intercalate into a WO_3_ crystal framework and occupy trigonal cavity sites [77,78]. We assume that if there is no regular layer structure for the modified WO_3_, lithium ions are not able to be inserted between the WO_3_ layers. Thus, the specific capacity of the hydrated and non-crystalline WO_3_ electrode material is much lower in comparison with the specific capacity of the bulk WO_3_-based anode material (Figure 10b), unlike for the measurements in an aqueous electrolyte, where the charge storage ability is mainly influenced by the pseudocapacity of the surface groups.

## 4. Conclusions

In the present work, the facile, eco-friendly water-based method is used to obtain hydrated and non-crystalline WO_3−x_. This method is described in the literature as an exfoliation process, but in our case, we cannot clearly confirm that the exfoliated material was obtained. The developed strategy leads to the formation of nanoflakes of the hydrated and non-crystalline WO_3_. Thus, it has been shown that the modified material is mainly amorphous, contains W(V) centers and is abundant with the surface groups. Bulk and non-crystalline WO_3_ were tested as electrode materials in both an aqueous and non-aqueous electrolyte. Modified WO_3_ exhibits a high specific pseudocapacitance due to the presence of surface groups, therefore, it can act as an efficient electrode material for supercapacitors. Multicyclic charge–discharge tests have confirmed the high electrochemical stability of the obtained material. On the other hand, a lack of crystallinity and a lack of a regularly layered structure do not allow the modified WO_3_ to participate in the lithium ion intercalation, thus it cannot be used as an efficient electrode material for lithium-ion batteries.

## Figures and Tables

**Figure 1 materials-13-01925-f001:**
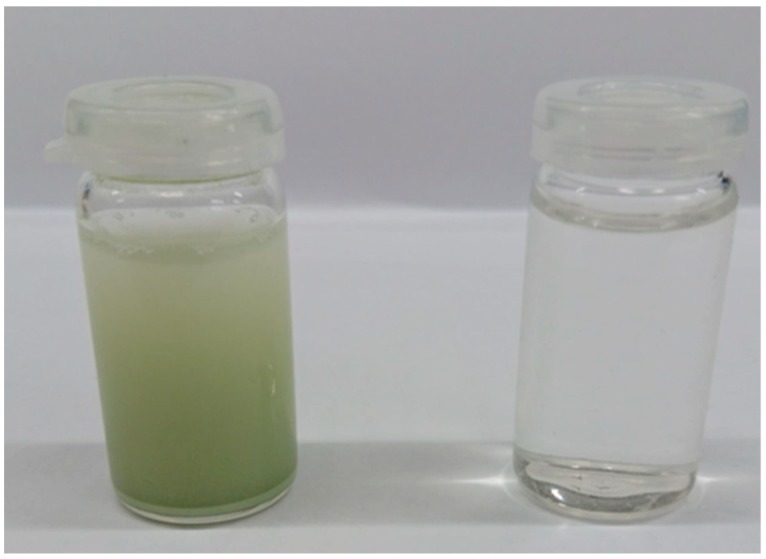
The photographs of WO_3_ suspensions before (left) and after (right) the exfoliation process.

**Figure 2 materials-13-01925-f002:**
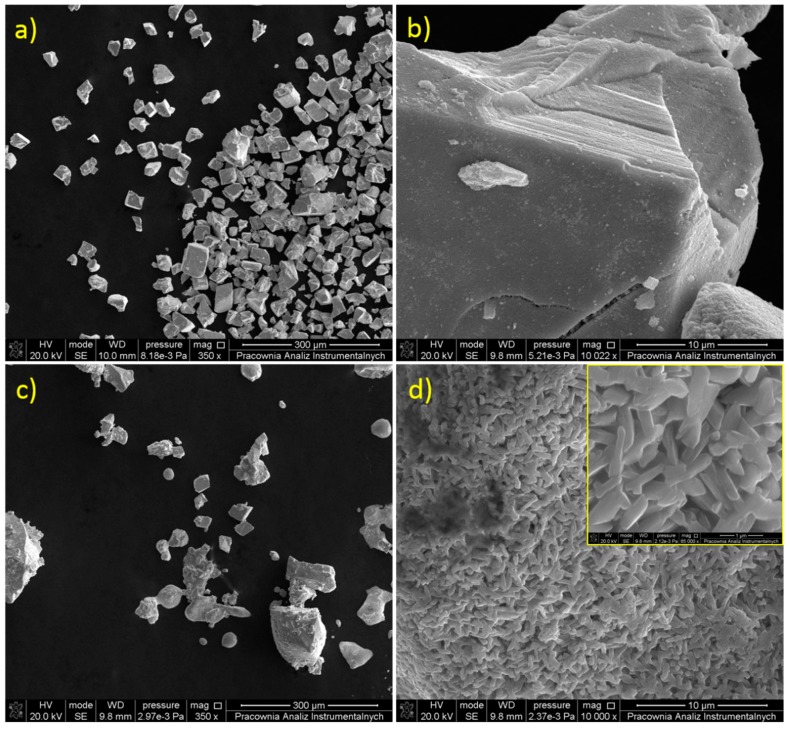
SEM images of (**a**,**b**) bulk and (**c**,**d**) hydrated and non-crystalline WO_3_.

**Figure 3 materials-13-01925-f003:**
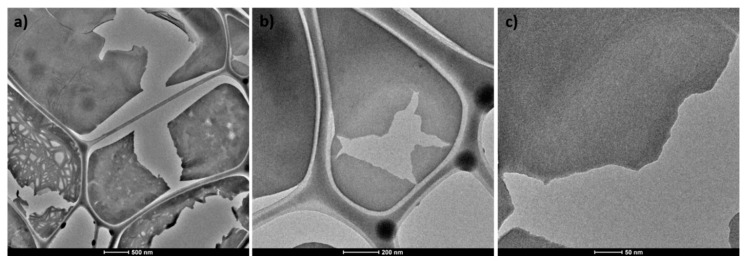
(**a**–**c**) TEM images of hydrated and non-crystalline WO_3−x_. at various magnifications.

**Figure 4 materials-13-01925-f004:**
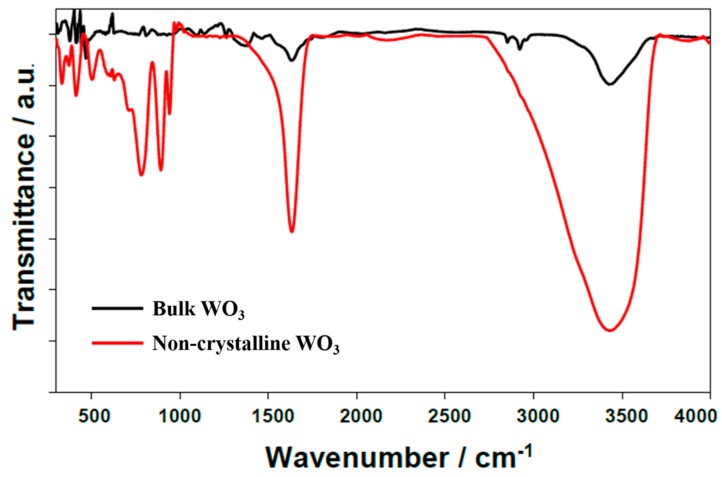
FTIR spectra of bulk and hydrated and non-crystalline WO_3_.

**Figure 5 materials-13-01925-f005:**
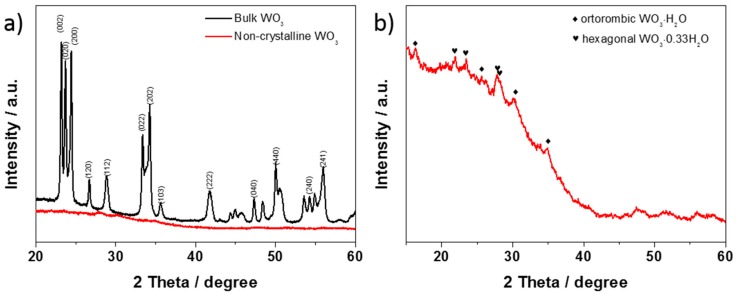
XRD patterns of (**a**) bulk and hydrated and non-crystalline WO_3_, (**b**) only hydrated and non-crystalline WO_3_.

**Figure 6 materials-13-01925-f006:**
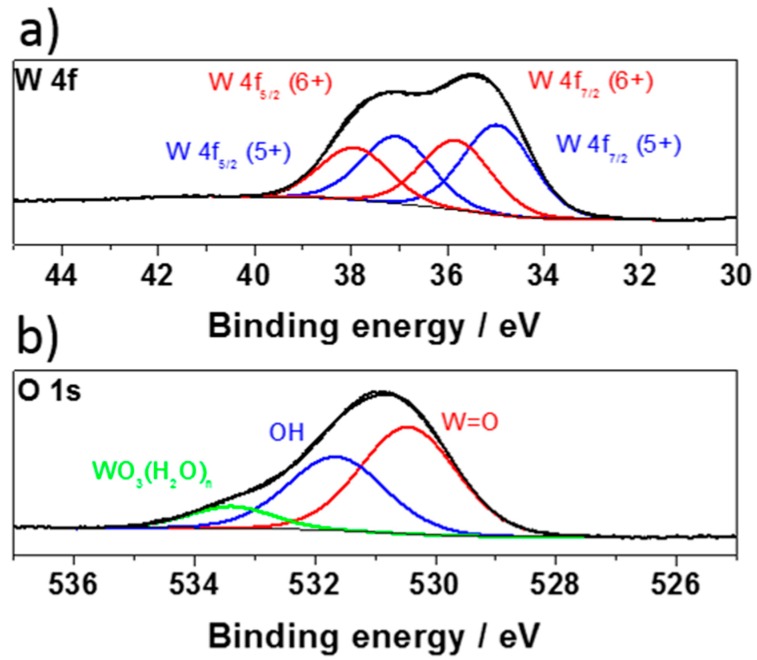
XPS spectra of (**a**) W 4f and (**b**) O 1S core levels of hydrated WO_3–x_.

**Figure 7 materials-13-01925-f007:**
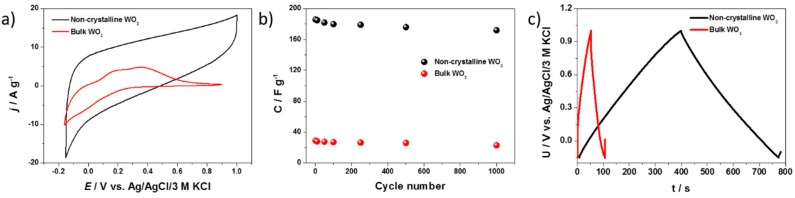
(**a**) Cyclic voltammograms for the non-crystalline and bulk WO_3_. (**b**) Curves of specific capacitance vs. cycle number for hydrated and bulk WO_3_. (**c**) Exemplary galvanostatic charge–discharge curves of electrodes at 0.2 A g^−1^ (in a three-electrode configuration).

**Figure 8 materials-13-01925-f008:**
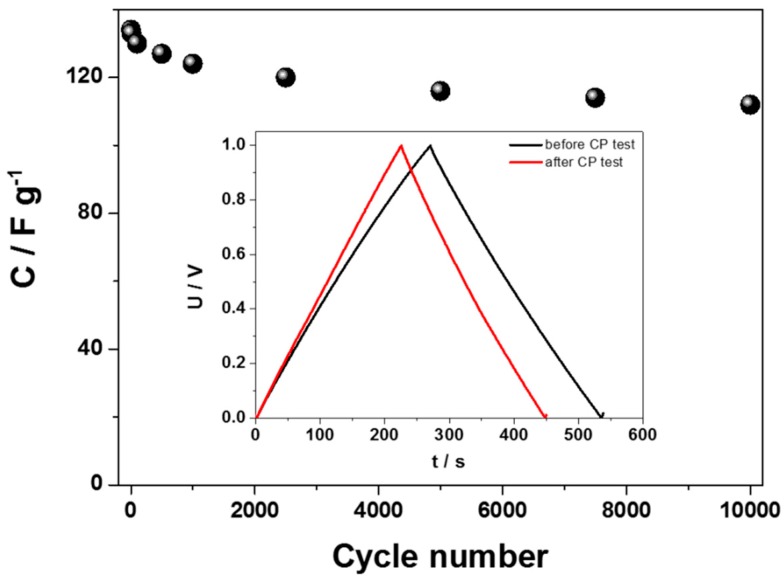
Curves of specific capacitance vs. cycle number for the hydrated non-crystalline WO_3−x_. Inset: exemplary galvanostatic charge–discharge curves of electrodes at 0.5 A g^−1^ (in a two-electrode configuration).

**Figure 9 materials-13-01925-f009:**
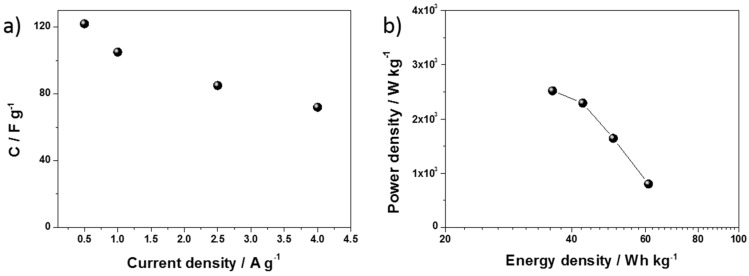
(**a**) Dependence of the specific capacitance of the hydrated WO_3−x_ electrode on the current density. (**b**) Ragone plot relating the energy and power densities of hydrated WO_3−x_ (in a two-electrode configuration).

**Figure 10 materials-13-01925-f010:**
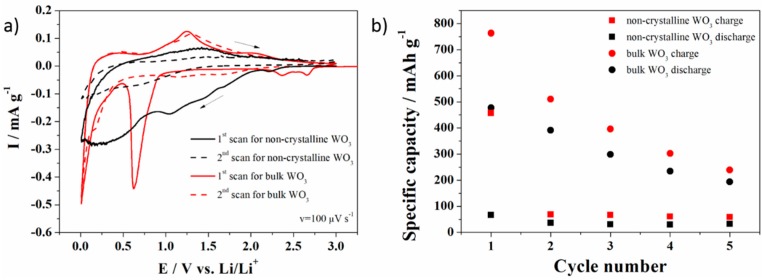
Electrochemical performance of the bulk and hydrated and non-crystalline WO_3_ electrode material in 1 M LiPF_6_ in EC/DMC 1:1 (**a**) cyclic voltammetry, (**b**) galvanostatic tests for battery application.

**Table 1 materials-13-01925-t001:** Methods and characteristics of the exfoliation process.

Material	Method	Precursor	Solvent	Layer Thickness	Crystalline Phase	Ref.
WO_3_ Nanosheets	Mechanical exfoliation	WO_3_∙2H_2_O	-	1.4–100	*m*-WO_3_	[53]
WO_3_ Nanosheets	Alcohothermal exfoliation	WO_3_∙*n*H_2_O	Absolute ethanol	8–20	*m*-WO_3_	[54]
Nano-WO_3_	Chemical exfoliation	Bi_2_W_2_O_9_	Concentrated hydrochloric acid; tetramethylammonium hydroxide solution	0.75	WO_3_∙0.5H_2_O	[55]
WO_3_ Nanosheets	Electrostatic-driven exfoliation	WO_3_ powder	BSA solution (pH 6-4)	1.4; 2.1	*m*-WO_3_	[56]
WO_3_∙2H_2_O Nanosheets	Ultrasonic exfoliation (intercalation with dodecylamine)	H_2_WO_4_	Concentrated nitric acid solution	1.4	WO_3_∙2H_2_O	[57]
WO_3_ Nanoflakes	Water exfoliation	Bulk WO_3_	Water	60–80	Amorphous	This work

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
