# Peer review of "An Easy and Ecological Method of Obtaining Hydrated and Non-Crystalline WO3−x for Application in Supercapacitors"

_materials, 2020, doi:10.3390/ma13081925_

Round 1

Reviewer 1 Report

The authors discuss the performances of exfoliated WO3 nanosheets as electrode materials in supercapacitors and LIBs. A high yield and green synthesis has been used to prepare the exfoliated WO3. Several techniques have been applied to put into evidence the peculiarities of the exfoliated WO3 with respect to the bulk one. The nanosheets display (i) very different morphology, (ii) loss of crystallinity, (iii) oxygen vacancies, adsorbed water and bound OH groups on the surface. The electrochemical investigation puts into evidence the higher specific capacitance in aqueous electrolyte and improved long-term cycling stability of the exfoliated WO3 with respect to the bulk one: the improvement is mainly related to the peculiarities reported in point (iii). On the contrary, worse electrochemical performances are obtained for exfoliated WO3 with respect to the bulk one when applied as anode in LIBs; it is explained by the observed amorphous phase and loss of the layered structure.  

The subject of the paper is appropriate for publication in “Materials”. The characterization results of the exfoliated WO3 explain the different electrochemical performances of the electrode when applied in supercapacitors and in LIBs. The reported results and discussion support the conclusions. Anyway, I suggest a minor revision of the paper by taking into account the following comments.

1) FTIR results

- Page 7 and Fig. 4. The peaks observed in the 600-1000 cm-1 range in the exfoliated sample are attributed to the O-W-O vibrations. They are not observed in the bulk sample. Any explanation for this?

3) Some typos are present.

Page 4, line 118: change “3 mgcm-1” into “3 mg cm-1

Page 7, line 179: change “band 3430 cm-1” into “band at 3430 cm-1

Author Response

Thank you for your remarks and suggestions.

The authors discuss the performances of exfoliated WO3 nanosheets as electrode materials in supercapacitors and LIBs. A high yield and green synthesis has been used to prepare the exfoliated WO3. Several techniques have been applied to put into evidence the peculiarities of the exfoliated WO3 with respect to the bulk one. The nanosheets display (i) very different morphology, (ii) loss of crystallinity, (iii) oxygen vacancies, adsorbed water and bound OH groups on the surface. The electrochemical investigation puts into evidence the higher specific capacitance in aqueous electrolyte and improved long-term cycling stability of the exfoliated WO3 with respect to the bulk one: the improvement is mainly related to the peculiarities reported in point (iii). On the contrary, worse electrochemical performances are obtained for exfoliated WO3 with respect to the bulk one when applied as anode in LIBs; it is explained by the observed amorphous phase and loss of the layered structure. 

The subject of the paper is appropriate for publication in “Materials”. The characterization results of the exfoliated WO3 explain the different electrochemical performances of the electrode when applied in supercapacitors and in LIBs. The reported results and discussion support the conclusions. Anyway, I suggest a minor revision of the paper by taking into account the following comments.

1) FTIR results

- Page 7 and Fig. 4. The peaks observed in the 600-1000 cm-1 range in the exfoliated sample are attributed to the O-W-O vibrations. They are not observed in the bulk sample. Any explanation for this?

Thank you very much for this comment

Added to the manuscript:

The W-O stretching modes are less intense, and changes in the low-frequency modes may indicate some modifications in the tungsten-oxide framework. The appearance of the most active surface centres suggests a connection with defects in nanoflakes [38]. The non-crystalline WO3 sample has a large number of defects that somehow activate W-O vibrations in range 600-1000 cm-1. This phenomenon is missing in the bulk sample, which is why the W-O vibrations are inactive, and thus not visible in FTIR spectrum are peaks 780, 890 and 940 cm-1. A similar effect was observed on the pristine Zn-Al sample and exfoliated [39].

[38]    S. Zhuiykov, E. Kats, Enhanced electrical properties in sub-10-nm WO3 nanoflakes prepared via a two-step sol-gel-exfoliation method, Nanoscale Res. Lett. 9 (2014) 1–10. doi:10.1186/1556-276X-9-401.

[39]    Y. Wei, F. Li, L. Liu, Liquid exfoliation of Zn-Al layered double hydroxide using NaOH/urea aqueous solution at low temperature, RSC Adv. 4 (2014) 18044-18051. doi:10.1039/c3ra46995f.

3) Some typos are present.

Page 4, line 118: change “3 mgcm-1” into “3 mg cm-1”

Page 7, line 179: change “band 3430 cm-1” into “band at 3430 cm-1”

Thank you for the comment. The corrections have been applied.

Reviewer 2 Report

General comments:

In my view, the experimental part of the article is entirely reliable and the experimental data set is extensive (the authors have worked a lot) but the interpretation of the experimental results seems forced: the authors presume that the material is exfoliated and hydrated, but there is more evidence of hydration (the FTIR are conclusive) that of exfoliation (neither the SEM nor the TEM micrographs provided show exfoliation nor does the XRD record provide a peak associated with any plane). The fact that an exfoliation method has been followed in the synthesis does not ensure that the product is exfoliated. Possibly, the exfoliated product has undergone hydration, which has determined its final properties. Please note that a simple heating in water produces only hydration of WO3, and a reducing agent would have been needed in the exfoliation to obtain crystalline WO3.

The term "hydrated" must be included in the title and references to lithium batteries should be avoided, since it creates expectations that do not correspond to reality. One thing is to value the results of the investigation and another is to assign properties which are not supported by the results.

Main issues:

Q1. Change the title of the article and include in the title "an easy and ecological method of obtaining hydrated and non-crystalline WO3-x". The title should not create expectations that the material can be used for lithium ion intercalation.
Q2. Suppress throughout the text "exfoliated WO3" and any references to the "formation of 2D nanolayers of hydrated WO3", and replace them with, for instance, "flakes of non-crystalline or amorphous WO3". It is important that only bulk WO3 and WO3 flakes (which are not crystalline and do not have a regular layered structure) are mentioned in the text.
Q3. The discussion needs to be adapted, i.e., it should be explained why WO3 bulk can be presented in 2D layers and how, from a simple exfoliation method, flakes of hydrated WO3 are obtained.
Q4. The text should discuss and leave no doubts about what exfoliation methods that actual lead to obtaining crystalline WO3, commenting on when and why amorphous WO3 materials are obtained, and when and why crystalline WO3 is obtained. A table with the existing bibliography, comparing them, and its discussion is a must.

Minor issues:

  • Lines 31-60 do not provide any useful information. They can be easily summarized in 10 lines, shortening the introduction.
  • L98: μL (capital L)
  • Figure 2: Exfoliation is not observed in (c) or (d).
  • Figure 3: These TEM micrographs do not show any exfolation.
  • L223: Add a space before "The"

Author Response

Thank you for your remarks and suggestions

In my view, the experimental part of the article is entirely reliable and the experimental data set is extensive (the authors have worked a lot) but the interpretation of the experimental results seems forced: the authors presume that the material is exfoliated and hydrated, but there is more evidence of hydration (the FTIR are conclusive) that of exfoliation (neither the SEM nor the TEM micrographs provided show exfoliation nor does the XRD record provide a peak associated with any plane). The fact that an exfoliation method has been followed in the synthesis does not ensure that the product is exfoliated. Possibly, the exfoliated product has undergone hydration, which has determined its final properties. Please note that a simple heating in water produces only hydration of WO3, and a reducing agent would have been needed in the exfoliation to obtain crystalline WO3.

Thank you very much to the reviewer for valuable information. We totally agree. The manuscript has been changed according to the reviewer's opinion.

The term "hydrated" must be included in the title and references to lithium batteries should be avoided, since it creates expectations that do not correspond to reality. One thing is to value the results of the investigation and another is to assign properties which are not supported by the results.

We totally agree. Thank you.

Main issues:

Q1. Change the title of the article and include in the title "an easy and ecological method of obtaining hydrated and non-crystalline WO3-x". The title should not create expectations that the material can be used for lithium ion intercalation.

A1. The authors thank you very much for the above remark and do agree that the previous title may have misled the reader. Therefore, the title has been changed.

Q2. Suppress throughout the text "exfoliated WO3" and any references to the "formation of 2D nanolayers of hydrated WO3", and replace them with, for instance, "flakes of non-crystalline or amorphous WO3". It is important that only bulk WO3 and WO3 flakes (which are not crystalline and do not have a regular layered structure) are mentioned in the text.

A2. The reviewer’s comments have been taken into account and the nomenclature has been standardized throughout the whole manuscript and the changes have been marked yellow.

Q3. The discussion needs to be adapted, i.e., it should be explained why WO3 bulk can be presented in 2D layers and how, from a simple exfoliation method, flakes of hydrated WO3 are obtained.

A3. Perhaps the exfoliation process, proposed in the literature and used by us, leads to obtaining exfoliated material, however the presence of obtained structures (see SEM images) results from the method of sample preparation that consists of slow evaporation of water which may partially reverse the exfoliation effect. Nevertheless, the proposed modification procedure clearly affects the morphology of WO3 and leads to loss of crystallinity of the material.

Add to the manuscript:

In this work, we used a process to modify the material, which is described in the literature as exfoliation. However, based on the results obtained, we cannot say unequivocally that we have obtained exfoliated material. Nevertheless, we received material characterized by non-crystallinity and the surface enriched in surface grups. Hence, this paper outlines the influence of a facile water-Based strategy and its effect on the structure, morphology and electrochemical properties of WO3. It is evidenced that this strategy may improve the electrode’s performance for a supercapacitor application but it is not convenient for the energy storage via faradaic reaction in batteries.

Perhaps, the exfoliation process, proposed in the literature and used by us, leads to obtaining an exfoliated material, however the presence of obtained, in this work, structures results from the method of electrode preparation that consists of slow evaporation of water which may partially reverse the exfoliation effect. Nevertheless, the proposed modification procedure clearly affects the morphology of WO3 and leads to the loss of crystallinity of the material.

Q4. The text should discuss and leave no doubts about what exfoliation methods that actual lead to obtaining crystalline WO3, commenting on when and why amorphous WO3 materials are obtained, and when and why crystalline WO3 is obtained. A table with the existing bibliography, comparing them, and its discussion is a must.

A4. The manuscript has been amended: the introduction part has been expanded to mention exfoliation methods and the application of amorphous materials. The part describing results and discussion has also been amended. Analyzing the literature results for WO3, collected in the Table 1, it can be stated that in almost every case of an exfoliation being applied, a crystalline, hydrated material is obtained. Attention was drawn to the fact that loss of crystallinity was observed when using aqueous exfoliation technique, which may be a link between both papers. Moreover, the exfoliated product might has undergone hydration, which has determined its final properties.

Changes in the manuscript:

New Introduction

The increase in global energy consumption, caused by the rapid increase in population, forces the production of energy storage and energy conversion systems on a larger scale. However, the utilization of fossil fuels has led to changes of the Earth’s climate by increased emission of the greenhouse gases [1]. Thus, new energy sources are expected to meet the requirements of being renewable and emission free [2]. To meet these expectations, more and more attention is being paid to energy storage devices such as supercapacitors and batteries. Supercapacitors may store electrical energy of high power density but low energy density without any conversion reactions [3]. Batteries require electrochemical reaction to obtain electrical energy from chemical reaction in a reversible way [4], assuming high energy density. Despite various energy storage mechanisms, one can assume that transition metal oxides may be successfully adapted as electrode materials for supercapacitors as well as for batteries. It is due to the fact that properties of metal oxides are strongly dependent on many factors i.e. crystal structure, morphology, doping and oxygen deficiency [5]. One of the method of some transition metal oxides modification is exfoliation that leads to 2D flakes formation.

Amusing physical and structural properties of atomically thin 2D materials are being the result of their improved surface-to-volume ratio, altered surface energy and confinement effect [6,7]. Over the last few years, rapid progress has been achieved in establishing procedures of graphene preparation [8] and transition metal dichalcogenides in a form of single or few layers [9–11] as a consequence of the slow breakdown of van der Waals force between adjacent layers in bulk. Nevertheless, the family of 2D materials is being expanded, especially by transition metal oxides to greatly develop their progressive applications [12].

Among many transition metal oxides, tungsten (VI) oxide (WO3) is attracting much attention. It has been used in electrochromic devices [13], gas sensors [14], electrocatalysis [15] and photoelectrocatalytic processes [16]. Recently, many studies have shown that WO3 may be utilized as energy storage material [17–19]. Lokhande et. al evidenced that crystal structure affects energy storage ability of WO3-based electrode materials [20]. There are many factors of synthesis that have an influence on the crystallographic structure, morphology and properties of obtained material i.e. temperature, pH, pressure, the time, the presence of additives etc. [21–23]. Exfoliation is one of the modification method to obtain 2D nanomaterials from layered materials [24], and is expected to improve electrode’s performance in terms of both stability and capacity [25].

Several approaches are accessible for the manufacture of WO3 2D materials, including hydrothermal, solvothermal, plasma treatment, acid etching, anodization and exfoliation. Over the last decade, many methods for the exfoliation of layered materials have been investigated for the synthesis of monolayer nanostructures. Mechanical exfoliation was firstly applied by Geim and Novoselov [26] to obtain graphene by using adhesive tapes. Nevertheless,  it is difficult to obtain uniform samples by this technique. The other one and one of the most commonly used techniques is liquid exfoliation that could be combined with oxidation, ion intercalation/exchange, as well as surface passivation by solvents [27]. In the case of metal oxides, the exfoliation usually involves sonication in surfactant solution and ion or polymer intercalation [28–30]. Application of the latter method leads to obtaining single layers in crystalline form, possessing the promising properties and gives large quantities of a dispersed nanomaterial. However, amorphous materials also find various applications, yet they are still gaining less interest. To our knowledge, most of the research in which amorphous materials of nanometric scale were obtained, are limited in zero-dimensional (0D) and one dimensional (1D), like amorphous MoO3-x nanoparticles or amorphous FeCoPOx nanowires [31–35]. All of the above mentioned showed excellent activity in catalysis and local surface plasmon resonances (LSPR). However, there are still many attempts to obtain 2D materials, especially with confined thickness,  that would have the superior activities and the ability of application in many fields. One of the latest studies, provided by Ren et al. [36], presented using the supercritical CO2 technology to obtain two-dimensional amorphous heterostructures of Ag/a-WO3-x and proposing the concept of synergistic photocatalysis that serves as a new methodology for the design of high-efficiency catalyst.

In this work, we used a process to modify the material, which is described in the literature as exfoliation. However, based on the results obtained, we cannot say unequivocally that we have obtained exfoliated material. Nevertheless, we received material characterized by hydration and non-crystallinity. Hence, this paper outlines the influence of a facile water-Based strategy and it’s effect on the structure, morphology and electrochemical properties of WO3. It is evidenced that this strategy may improve the electrode’s performance for a supercapacitor application but it is not convenient for the energy storage via faradaic reaction in batteries.

Perhaps, the exfoliation process, proposed in the literature and used by us, leads to obtaining an exfoliated material, however the presence of obtained, in this work, structures results from the method of sample preparation that consists of slow evaporation of water which may partially reverse the exfoliation effect. Nevertheless, the proposed modification procedure clearly affects the morphology of WO3 and leads to the loss of crystallinity of the material.

Results and discussion:

Due to the fact that the obtained result in the form of hydrated and non-crystalline material was unexpected after applying the exfoliation technique, it is difficult to make a comparison as most of the exfoliation results provide a crystalline phase of material. What is more, the process of exfoliation of transition metal oxides, a tungsten oxide in particular, is not as widely studied as, for example, the process of graphene exfoliation. However, there are individual studies on this topic. In both Kalantar-Zadeh et. al [41] and Yan et. al [42] used hydrated WO3 as the precursor, which is a typical layered material consisting of planes connected through the interaction of oxygen and hydrogen through hydrogen bonds in adjacent layers. Waller et. al [43] exfoliated Bi2W2O9 and in each of these research, the crystalline phase of the final product was obtained. Comparing the results presented in Table 1 it can be stated that regardless of which exfoliation technique was used, tungsten oxide was obtained in crystalline form. In the case of a nanoflakes obtained during the experiment presented in this work, the exfoliated product might has undergone hydration, which has determined its final properties. The use of water exfoliation technique may also be related to obtaining poor crystallinity. Similar results were obtained by Etman et al [44], who also observed poor crystallinity for MoO3 nanosheets, also utilizing water-based exfoliation..

Table 1. Methods and characteristics of the exfoliation process.

Material

Method

Precursor

Solvent

Layer thickness [nm]

Crystalline phase

Ref.

WO3 Nanosheets

Mechanical exfoliation;

WO3∙2H2O

-

1.4-100

m-WO3

[12]

WO3 Nanosheets

Alcohothermal exfoliation;

WO3nH2O

Absolute ethanol

8-20

m-WO3

[13]

Nano- WO3

Chemical exfoliation

Bi2W2O9

Concentrated hydrochloric acid; tetramethylammonium hydroxide solution

0.75

WO3∙0.5H2O

[14]

WO3 Nanosheets

Electrostatic-driven exfoliation

WO3 powder

BSA solution (pH 6-4)

1.4; 2.1

m-WO3

[16]

WO3∙2H2O Nanosheets

Ultrasonic exfoliation (intercalation with dodecylamine)

H2WO4

Concentrated nitric acid solution

1.4

WO3∙2H2O

[17]

WO3 Nanoflakes

Water exfoliation

Bulk WO3

Water

60-80

Amorphous; WO3∙H2O; WO3∙0.33H2O

This work

Minor issues:

Lines 31-60 do not provide any useful information. They can be easily summarized in 10 lines, shortening the introduction.

L98: μL (capital L)

Figure 2: Exfoliation is not observed in (c) or (d).

Figure 3: These TEM micrographs do not show any exfolation.

L223: Add a space before "The"

The authors thank you very much for the above remarks.

Figure captions have been changed.

Large corrections were also made for the Introduction.

Reviewer 3 Report

This manuscript entitled “Exfoliated WO3 nanosheets as an electrode material for supercapacitors and an anode for lithium-ion batteries” manuscript no. Materials-766392, discuss a systematic synthesis of hydrated exfoliated WO3 nanosheets (WO3-x) via an environmentally friendly aqueous exfoliation approach which exhibits 7 times higher specific capacitance than bulk WO3 and demonstrates an outstanding long-term cycling stability. Such study can provide significant information to readers about the exfoliation effect on transition metal oxides towards charge storage. This work is publishable on Materials after a major revision by fully considering the following suggestions.

  1. Authors should discuss in detail for utilizing tungsten (VI) oxide (WO3), although very cheap Mn, Ni or Co based metal oxides are reported to demonstrate far better charge storage performance than the present report. Author should compare the results in tabular form with other similar materials reported for charge storage device.
  2. Author has discussed the thickness of flakes to be 50-150 nm based on SEM micrograph shown in inset Fig 3D. This micrograph is of scale 1mm and is not correct for evaluation of thickness of order 50 nm. Kindly include cross-sectional SEM on higher magnification.
  3. Author should include adsorption/desorption isotherms to examine the increase in surface area after exfoliation of WO3. Determine the pore-size distribution and should relate the electrolytic ionic diffusion through these pores and thus explain their charge storage in both aqueous and non-aqueous mediums.
  4. Perform scan dependent CV from 2 mV/s to 500 mV/s and then evaluate whether charge storage is diffusion control or surface control. Using these CV evaluate the capacitive retention.
  5. Discuss in experimental section, whether the GCD is performed using 2-electrode configuration. If not, then author should perform all measurements in 2-electrode configuration to evaluate the device performance.
  6. GCD at different current density is necessary and a Ragone plot should be shown compared to other literature reports. Csp can be calculated by integration method as discussed in Applied Surface Science, 2018, Volume 427, Pages 102-111, doi.org/10.1016/j.apsusc.2017.08.028. The Energy density and power densities should be evaluated and compared in Ragone plot.
  7. GCD Figure 7C, displays triangular-shaped CD profiles, why such triangular or symmetric shape is observed, although the electrode material is just pseudocapacitive metal-oxide?
  8. GCD cycling should be performed atleast 10,000 cycles to make some conclusive decision on the stability of the fabricated device.
  9. Electrochemical Impedance Spectroscopy (EIS) measurements should be conducted to examine the quantitative resistive and capacitive components. This can be evidenced by the reduction of charge transfer impedance. From EIS results Diffusion coefficient should be derived. Authors can refer following references for the same: Electrochimica Acta, 2015, Volume 169, Pages 276-282, doi.org/10.1016/j.electacta.2015.03.141; J. Mater. Chem. A, 2015, Volume 3, Pages 9925-9931, doi:10.1039/C5TA00653H; Applied Surface Science, 2018, Volume 427, Pages 102-111, doi.org/10.1016/j.apsusc.2017.08.028; Applied Surface Science, 2017, Volume 404, Pages 197-205, doi.org/10.1016/j.apsusc.2017.01.300; Materials Research Bulletin, 2016, Volume 83, Pages 167-171, doi.org/10.1016/j.materresbull.2016.06.006; PloS one, 2015, Volume 10, Pages e0131475, doi.org/10.1371/journal.pone.0131475.
  10. Author claims: Line 236 “an increased contact surface of the material with an electrolyte due to the 2D structure”. Author should provide contact angle measurement of electrolyte drops on both electrodes in order to ascertain this claim.
  11. The headings Aqueous electrolyte and Non-aqueous electrolyte is not suitable for the discussed content below. Kindly change it.
  12. How non-reversible behavior of electrodes in Non-aqueous electrolyte (Figure 8A) can be useful for anode material in lithium-ion batteries?
  13. Why there is rapid drop of specific capacity in Figure 8B in just 5 cycles and how such materials can be used for practical purpose?
  14. Heading of this article includes lithium-ion batteries, so Author should perform more tests required for battery performance and stability.
  15. Few typo errors should be removed: Line 191: “halo, etc.

Good Luck.

Author Response

Thank you for your remarks and suggestions.

This manuscript entitled “Exfoliated WO3 nanosheets as an electrode material for supercapacitors and an anode for lithium-ion batteries” manuscript no. Materials-766392, discuss a systematic synthesis of hydrated exfoliated WO3 nanosheets (WO3-x) via an environmentally friendly aqueous exfoliation approach which exhibits 7 times higher specific capacitance than bulk WO3 and demonstrates an outstanding long-term cycling stability. Such study can provide significant information to readers about the exfoliation effect on transition metal oxides towards charge storage. This work is publishable on Materials after a major revision by fully considering the following suggestions.

Q1. Authors should discuss in detail for utilizing tungsten (VI) oxide (WO3), although very cheap Mn, Ni or Co based metal oxides are reported to demonstrate far better charge storage performance than the present report. Author should compare the results in tabular form with other similar materials reported for charge storage device.

A1. According to the above remarks, information about some other metal oxides has been included in the introduction part. Moreover, comparison of the others materials used in charge storage devices was described in detail in answer to question Q6. The possible applications of tungsten oxide and its properties had been already mentioned in the introduction.

Add to the manuscript:

Apart from tungsten oxide, there is a major number of other metal oxides that were reported to enhance working properties of energy storage devices. Ruthenium oxide is widely investigated because of its high specific capacitance (up to 700 F/g), but its application is severely limited by the high cost [29]. Metal oxides such as MnO2 [30] or NiO [31] have similar advantages and there have been some attempts to apply as electrode materials. However, their poor electrical conductivity affects the speed of charging and contributes to severe capacitance loss. To improve the capacitive performance of materials with poor electrical conductivity, many researchers found oxygen-defective metal oxides to be of the greatest interest. Reported results show that the concentration of oxygen vacancies has significant influence on the structure, as well as charge storage properties, thus enable an excellent cycling performance [32–35].

Q2. Author has discussed the thickness of flakes to be 50-150 nm based on SEM micrograph shown in inset Fig 3D. This micrograph is of scale 1mm and is not correct for evaluation of thickness of order 50 nm. Kindly include cross-sectional SEM on higher magnification.

A2. We agree with the reviewer's opinion. We tried to take an SEM on higher magnification, but it was difficult to get the micrograph with satisfactorily sharpness. Nevertheless, based on the SEM micrograph in Fig. S1, we can roughly estimate the thickness to be in order of 50-100 nm.  

Fig. S1. SEM on higher magnification for hydrated WO3-x.

changes in the manuscript:

Aggregates of exfoliated WO3 are built of randomly oriented plates of heterogeneous shapes with a size of 0.7-1.5 μm, see Fig 2d and inset Fig. 2d. The thicknesses of the flakes were roughly estimated to be in the order of 50 - 100 nm, see Fig. S1 in Supplementary Information.

Q3. Author should include adsorption/desorption isotherms to examine the increase in surface area after exfoliation of WO3. Determine the pore-size distribution and should relate the electrolytic ionic diffusion through these pores and thus explain their charge storage in both aqueous and non-aqueous mediums.

A3. We agree with the Reviewer that the comparison of surface area of bulk and hydrated material should be performed We are very sorry, but we are not able to make these measurements, because Faculty of Applied Physics and Mathematics, where such measurements can be investigated, is closed due to COVID-19. Nevertheless, we would like to point out that all electrochemical measurements, that could be performed in our lab, have been done according to the comments.

Q4. Perform scan dependent CV from 2 mV/s to 500 mV/s and then evaluate whether charge storage is diffusion control or surface control. Using these CV evaluate the capacitive retention.

A4. Thank you for this comment. As it was expected, charge storage on hydrated WO3-x electrode material is controlled by surface processes.

added to the manuscript:

The CVs of hydrated WO3-x recorded at different scan rates (5-500 mV s-1) are presented in Fig. S2. The anodic current density at 0.5 V shows a linear relation with the scan rate (see Fig. S2b), suggesting that charge storage is controlled by surface processes, thus it is not diffusion-controlled phenomenon, see Fig.S2c.

Fig. S2. a) CV curves of hydrated WO3-x electrode in 0.2 M K2SO4. Scan rates 5–500 mV s-1. Dependence of anodic current at 0.5 V b) vs. scan rate and c) vs. square root of the scan rate.

Q5. Discuss in experimental section, whether the GCD is performed using 2-electrode configuration. If not, then author should perform all measurements in 2-electrode configuration to evaluate the device performance.

A5. Thank you for this comment. In the previous version of the manuscript, GCD measurements were only performed using a 3-electrode configuration. In the new version of this work, we present the GCD measurements for WO3 in a 2-electrode, symmetric configuration.

added to the manuscript

Experimental part:

Galvanostatic charge/discharge measurements for hydrated WO3-x were also performed using fully assembled symmetric two-electrode cells in a coffee bag system. The commercially available foil was used for the preparation of the supercapacitor cells. A Whatman paper was used as a separator. An aqueous solution of 0.2 M K2SO4 was used as an electrolyte. Coffee bags were enclosed under a vacuum using a Mini Jumbo Henkelman Vacuum System. The supercapacitor cells were tested using multiple galvanostatic charge/discharge cycles (10000 cycles, jc = ja = 0.5 A g-1).

Results and discussion part:

Multiple charge/discharge cycles in a two-electrode configuration for hydrated WO3-x.were performed in order to test the stability of the tested supercapacitor (see Fig. 8). As it is shown, a very good stability, even after 10,000 cycles, was obtained for the hydrated WO3-x based capacitor. The capacitance retention between the 1st and 10,000th chronopotentiometry cycle was equal to 84%. The effect of the capacitance drop was also tracked using chronopotentiometry. The curves recorded before and after a long-term test are presented also in Fig. 8 inset. It is noteworthy that the decrease of the capacitance is the highest at the beginning of the charge/discharge tests, and then the capacitance stabilized after approx. 2000 cycles. This means that the capacitance drop is not related to the electrolyte decomposition, but some irreversible reactions on the material surface.

Fig. 8. Curves of specific capacitance vs. cycle number for hydrated non-crystalline WO3-x. Inset: exemplary galvanostatic charge/discharge curves of electrodes at 0.5 A g-1 (in two-electrode configuration).

Q6. GCD at different current density is necessary and a Ragone plot should be shown compared to other literature reports. Csp can be calculated by integration method as discussed in Applied Surface Science, 2018, Volume 427, Pages 102-111, doi.org/10.1016/j.apsusc.2017.08.028. The Energy density and power densities should be evaluated and compared in Ragone plot.

A6. Thank you for this comment.

added to the manuscript:

The dependence of the specific capacitance of the hydrated WO3-x electrode on current density is shown in Fig. 9a. The specific capacitance (Cs) was 122 A g-1 and 72 A g-1 at current densities of 0.5 A g-1 and 4 A g-1, respectively. The capacitances were found to decrease by increasing the charge/discharge cell current. This is because, at higher current densities, the slower processes demonstrate a kinetic resistance and cannot participate in a charge transfer onto or across the electrode/ electrolyte interface.

The Ragone plots display the relationship between power density and energy density. In the case of hydrated WO3-x, at a power density of 803 W kg-1, an energy density of 60 W h kg-1 was obtained. When power density increased to 2520 W kg-1, energy density was 36 Wh kg-1 (see Fig. 9b), which is quite impressive as compared to earlier reports about modified metals oxides (Table S1) [55–61].

Fig. 9. a) Dependence of the specific capacitance of hydrated WO3-x electrode on current density. b) Ragone plot relating energy and power density of hydrated WO3-x (in two-electrode configuration).

Table S1. Comparison of the electrochemical properties of the non-crystalline WO3 with some previous reports on metals oxide-based supercapacitors

Electrode material

Cs [F g-1]

Current density or scane rate

Energy Density

[Wh kg-1]

Power Density

[W kg-1]

Ref.

hexagonal WO3

484

0.93 A/g

25

89

[68]

Graphene-TiO2

165

5 mV/s

12.5

1440

[69]

MnO2-RuO2@GNR

156

1 A/g

60

14000

[70]

Cu2O/CuMoO4 nanosheets

156

1 A/g

75.1

420

[71]

Pd doped monoclinic WO3

41

0.5 A/g

10.6

198

[72]

Graphene-WO3 Nanowires

465

1 A/g

26.7

6000

[73]

WO3-MnO2

103

5 mV/s

24.13

915

[74]

Hydrated WO3-x

122

0.5 A/g

60

803

This work

Q7. GCD Figure 7C, displays triangular-shaped CD profiles, why such triangular or symmetric shape is observed, although the electrode material is just pseudocapacitive metal-oxide?

A7. We would like to thank the Reviewer very much for this comment. It forced us to think about the mechanism of energy storage by our material. It is known that pseudocapacitive material is characterized by the presence of cathodic and anodic maxima due to electrode material reduction and oxidation, respectively (B.K. Kim, S. Sy, A. Yu, J. Ahang, Electrochemical Supercapacitors for Energy Storage and Conversion in Handbook of Clean Energy Systems in 2015 by John Wiley & Sons, Ltd). The Cv of the bulk WO3 exhibits typical pseudocapacitive character while cv of  hydrated and non-crystalline WO3 does not exhibit maxima (Fig. 7A). However, we suppose that the both, pseudocapacitance (PC) as well as electrical double layer capacitance (EDLC) contribute to overall energy storage mechanism of hydrated WO3-x. The lack of clear maxima on CV and plateau on GCD  may be related to the lack of material crystallinity. No ordered structure leads to the pseudocapacitance activity extension to the whole tested potential range. It was previously reported that WO3-based supercapacitor exhibited both PC and EDLC, however, the contribution of pseudocapacitance was much higher. In the case of hydrated WO3-x, electrode material characterized by the 2D structure as well as the surface enriched in surface groups, it may be expected the contribution of EDLC is significantly higher in comparison with bulk WO3. The shape of Cv and GCD curves is affected by both, EDLC and pseudocapacitance. However, the determination of the quantitative contribution of them would require additional investigation. The short discussion has been added to the manuscript.

Q8. GCD cycling should be performed atleast 10,000 cycles to make some conclusive decision on the stability of the fabricated device.

A8. Thank you for this comment. To test the stability of the supercapacitor, we performed 10,000 GCD cycles. The result is presented above (the reviewer's question number 5).

Q9. Electrochemical Impedance Spectroscopy (EIS) measurements should be conducted to examine the quantitative resistive and capacitive components. This can be evidenced by the reduction of charge transfer impedance. From EIS results Diffusion coefficient should be derived. Authors can refer following references for the same: Electrochimica Acta, 2015, Volume 169, Pages 276-282, doi.org/10.1016/j.electacta.2015.03.141; J. Mater. Chem. A, 2015, Volume 3, Pages 9925-9931, doi:10.1039/C5TA00653H; Applied Surface Science, 2018, Volume 427, Pages 102-111, doi.org/10.1016/j.apsusc.2017.08.028; Applied Surface Science, 2017, Volume 404, Pages 197-205, doi.org/10.1016/j.apsusc.2017.01.300; Materials Research Bulletin, 2016, Volume 83, Pages 167-171, doi.org/10.1016/j.materresbull.2016.06.006; PloS one, 2015, Volume 10, Pages e0131475, doi.org/10.1371/journal.pone.0131475.

A9. According to the Reviewer`s suggestion, we have performed EIS measurements in aqueous electrolyte of both electrode materials deposited on FTO. Two spectra for each type of electrode was registered. The first one was recorded close to the rest potential (0 V vs Ag/AgCl (3 M KCl), and the second under anodic polarization (0.6 V vs Ag/AgCl (3 M KCl). The comparison of spectra is shown below. Three of four models were fitted to data using the same equivalent circuit that consists: R1 – electrolyte resistance, R2 – resistance on the electrode/electrolyte interface, R3 – bulk resistance, CPE1 – constant phase element on the electrode/electrolyte interface, and CPE2 – constant phase element in bulk (EQC 1). In the case of the spectrum of bulk WO3 recorded at a rest potential, additional Warburg element has to be added in order to fit model properly (EQC 2).

The Warburg element was necessary only in the case of the spectrum of bulk material recorded at 0 V. As it can be seen on the spectrum, a straight line inclined at an angle of 45 degrees at the lowest frequencies characteristic for diffusion was recorded. Noteworthy, the spectrum of hydrated WO3-x electrode material does not exhibit this behavior, and can be simply modeled using EQC1. It is in a good agreement with results presented in the manuscript. The diffusion of ions into hydrated WO3-x is not expected. Due to the fact that diffusion processes were recorded for only one spectrum, diffusion coefficients cannot be compared. The fit parameters, presented below, are affected by the electrochemical process that occurs at certain potential.

at 0 V

bulk

hydrated WO3-x

R1 / Ω

23.15

26.01

R2 / Ω

336.82

1.70E+05

R3 / Ω

254.54

3966.50

P1/ Ω-1sn

1.63E-05

9.99E-06

n1

1

0.95

P2/ Ω-1sn

0.00011

4.38E-05

n2

0.77

0.99

Wor / Ωs-0.5

1466.40

Woc / s0.5

7.92

The spectra were also recorded under anodic polarization at steady-state conditions (current did not exceed 300 nA) and obtained data fulfil Kramers–Kronig test. The spectra of hydrated WO3-x exhibits typical capacitive behavior. The fit parameters are compared in a form of a table.

at 0.6 V

bulk

hydrated WO3-x

R1 / Ω

24.00

25.72

R2 / Ω

1.53E+05

3.60E+06

R3 / Ω

1.08E+04

4.66

P1 / Ω-1sn

8.05E-06

5.54E-06

n1

0.99

0.98

P2/ Ω-1sn

1.55E-05

1.39E-04

n2

0.93

0.83

The most significant differences are seen in the values of R2, R3, and P2 (from CPE2). It can be concluded that resistance on the electrode/electrolyte interface is higher for modified WO3, however, the resistance within electrode material is 4 order of magnitude lower for hydrated WO3-x. The P2 parameter of CPE2 (in the case of n close to 1) mainly contains a capacitive component. The one order of magnitude higher value was obtained for material after proposed modification. Thus, EIS analysis confirms that hydrated WO3-x material can act as an electrode material for energy storage devices, mainly supercapacitors.

Changes in the manuscript:

In the experimental part:

The electrochemical impedance spectra (EIS) for both electrode materials were recorded using AutoLab PGStat10 for the working electrode at its rest potential and under anodic polarization. The frequency range covered 10 kHz – 0.32 Hz (90 points), whereas the amplitude of the AC signal equaled 10 mV. The following elements were used for the fitting procedure of measured impedance spectra using EIS Analyzer software: R – resistance, CPE – constant phase element, and ZWofinite length diffusion impedance, where:

 and .                    

In the Electrochemical properties part:

Both electrode materials were compared using electrochemical impedance spectroscopy. Two spectra for each type of electrode were registered (Fig. S4). The first one was recorded close to the rest potential (0 V vs. Ag/AgCl (3 M KCl), and the second under anodic polarization (0.6 V vs. Ag/AgCl (3 M KCl). The results, equivalent circuits, and fitting parameters are presented in the supplementary information, see Fig. S3 and Table S1. The EIS analysis confirms that the diffusion process affects the impedance spectrum only of bulk WO3 recorded at 0 V, seen as a straight line inclined at an angle of 45º at the lowest frequencies, and the Warburg element (Wo) is necessary to fit model properly. In the case of a hydrated WO3-x model can be fitted using a simpler equivalent circuit, without Wo. The comparison of fitting parameters of bulk and hydrated tungsten oxide electrode recorded under anodic polarization confirms that hydrated WO3-x can act as electrode material for energy storage devices, mainly supercapacitors.

In supplementary information:

Three of four models were fitted to data using the same equivalent circuit that consists: R1 – electrolyte resistance, R2 – resistance on the electrode/electrolyte interface, R3 – bulk resistance, CPE1 – constant phase element on the electrode/electrolyte interface, and CPE2 – constant phase element in bulk (EQC 1). In the case of the spectrum of bulk WO3 recorded at a rest potential, additional Warburg element has to be added in order to fit model properly (EQC 2).

Fig. S3 The equivalent circuits used for modeling.

Fig. S4 The impedance spectra, experimental and fitted, of bulk WO3 and hydrated WO3-x recorded at 0 V and 0.6 V vs. Ag/AgCl (3 M KCl).

In the case of measurements performed under 0.6 V, the spectra of hydrated WO3-x exhibits typical capacitive behavior. The most significant differences of fitting parameters between electrodes under anodic polarization are seen in the values of R2, R3, and P2 (from CPE2). It can be concluded that resistance on the electrode/electrolyte interface is higher for modified WO3, however, the resistance within electrode material is 4 order of magnitude lower for hydrated WO3-x. The P2 parameter of CPE2 (in the case of n close to 1) mainly contains a capacitive component. The one order of magnitude higher value was obtained for material after the proposed modification. Thus, EIS analysis confirms that hydrated WO3-x material can act as an electrode material for supercapacitors.

Table S1 The results of fitting procedure.

at 0.6 V

at 0 V

bulk

hydrated WO3-x

bulk

hydrated WO3-x

R1 / Ω

24.00

25.72

R1 / Ω

23.15

26.01

R2 / Ω

1.53E+05

3.60E+06

R2 / Ω

336.82

1.70E+05

R3 / Ω

1.08E+04

4.66

R3 / Ω

254.54

3966.50

P1 / Ω-1sn

8.05E-06

5.54E-06

P1/ Ω-1sn

1.63E-05

9.99E-06

n1

0.99

0.98

n1

1

0.95

P2/ Ω-1sn

1.55E-05

1.39E-04

P2/ Ω-1sn

0.00011

4.38E-05

n2

0.93

0.83

n2

0.77

0.99

Wor / Ωs-0.5

1466.40

Woc / s0.5

7.92

Q10.Author claims: Line 236 “an increased contact surface of the material with an electrolyte due to the 2D structure”. Author should provide contact angle measurement of electrolyte drops on both electrodes in order to ascertain this claim.

A10. This sentence has been deleted because, as suggested by Reviewer 2, it does not match the new version of the manuscript. Moreover, the same as in the case of BET measurements, we are not able to make these measurements, due to the COVID-19 epidemic that limits access to laboratory equipment.

Q11. The headings Aqueous electrolyte and Non-aqueous electrolyte is not suitable for the discussed content below. Kindly change it.

A11. Unfortunately, we do not understand this comment. We think the chapters are well signed. The main chapter 3.2 is entitled Electrochemical properties, and then 3.2.1 “Aqueous electrolyte”. This seems to be a logical division..

Q12. How non-reversible behavior of electrodes in Non-aqueous electrolyte (Figure 8A) can be useful for anode material in lithium-ion batteries?

A12. In Figure 10 we have shown the behaviour of bulk and non-crystalline WO3 in non-aqueous electrolyte. We do not claim that non-reversible behaviour of electrode in any energy storage system is of advantage. If electrode material exhibits high irreversible capacity in the first cycle its utilization in commercial batteries is very doubtful. We agree with Reviewer that any electrode material should exhibit very low irreversible capacity loss in the first and next cycles.

Q13. Why there is rapid drop of specific capacity in Figure 8B in just 5 cycles and how such materials can be used for practical purpose?

A13. We wish to have a more stable material after modification. In general, our procedure for any material for battery application is consisted on performing 5 cycles at very low current density. We apply current is equal to C/20-rate (it means that in 20 hours electrode material should be charged or discharged). Commercially available graphite as an anode material exhibits the theoretical specific capacity of 372 mAh/g. As it is shown in Fig. 10B, bulk WO3 reached ~ 280 mAh/g while the specific capacity of exfoliated WO3 was below 100 mAh/g.
We do not claim that 5 cycles is enough to state that material is suitable as anode/cathode material. We just wanted to show that nor bulk WO3 neither non-crystalline WO3 are promising negative electrode material for lithium-ion battery application in our study.

Q14. Heading of this article includes lithium-ion batteries, so Author should perform more tests required for battery performance and stability.

A14. Thank you for that comment. We see no point to perform more tests as it has been already shown that for low current density material exhibited very low value of specific capacity. If higher current densities are applied the obtained specific capacities will be much worse.
However, we decided to change title of the manuscript as the old title might have been misleading for readers (modified WO3 is not recommended to be used as an electrode material for lithium-ion batteries).

Q15. Few typo errors should be removed: Line 191: “halo, etc.

A15. Thank you for the comment. The corrections have been applied.

Good Luck.

Thank You ?

Round 2

Reviewer 2 Report

Thank you for thoroughly addressing all the issues raised in previous iteration.

Reviewer 3 Report

Authors have done sufficiently good experiments to improve the manuscript and express results more clearly.

It can be recommended for publication.

Hope you and your team are doing good to cope with the present condition of COVID-19 crisis!

Good Luck